# The burden of HIV/AIDS in Ethiopia: Unveiling 30 years of trends in incidence, mortality, and disability—insights from the Global Burden of Disease study (1990–2021)

**Tegene Atamenta Kitaw**[1]*, **Molla Azmeraw**[1], **Biruk Beletew Abate**[1,3],
**Befkad Derese Tilahun**[1], **Alemu Birara Zemariam**[1], **Addis Wondmagegn Alamaw**[1],
**Abebe Merchaw Faris**[1], **Addisu Getie**[2], **Molalign Aligaz Adisu**[1], **Tesfaye Engdaw Habtie**[1],
**Melesse Abiye Munie**[2], **Ribka Nigatu Haile**[1]

1 Department of Nursing, College of Health Science, Woldia University, Woldia, Ethiopia, 2 Department of Nursing, College of Medicine and Health Sciences, Debre Markos University, Debre Markos, Ethiopia, 3 School of Population Health, Curtin University, Perth, WA, Australia

* tegene2013@gmail.com

## Abstract

### Background

Over the past three decades, Ethiopia has witnessed a dynamic shift in the burden of HIV/AIDS, a public health crisis that has profoundly impacted communities nationwide. While significant strides have been made in combating the disease, evolving trends in incidence, mortality, and disability reveal the complexity of progress. The insights drawn from the Global Burden of Disease Study (1990–2021) offer a comprehensive perspective on the successes and ongoing challenges. Understanding these trends is critical for refining public health strategies and achieving more equitable health outcomes. This study delves into the data, unveiling key findings that inform future policies and interventions.

### Methods

This study used data from the 2021 Global Burden of Disease Study (1990–2021) to evaluate HIV/AIDS incidence, mortality, and DALYs in Ethiopia. The dataset, including age-standardized rates and 95% uncertainty intervals, was obtained from the Global Health Data Exchange (GHDx). Statistical analyses included Joinpoint regression to identify trend shifts and calculate the Annual and Average Annual Percentage Change over time, with significance determined at a p-value of 0.05. Geographic variations were visualized using ArcGIS Pro, highlighting regions with varying HIV burdens. The comprehensive methodology adhered to GATHER guidelines for transparent reporting.

### Results

In 2021, 715,839 individuals (95% UI: 568,889.58–914,511.59) were living with HIV in Ethiopia, and 839,819.31 DALYs (95% UI: 620,558.56–1,154,483.92) were recorded. There was a significant decline in the age-standardized HIV incidence rate, from 175.4

**Data availability statement:** For access to the data utilized in this analysis, please refer to the GBD 2021 data-input sources tool available on the Global Health Data Exchange (GHDx) platform at https://ghdx.healthdata.org/gbd-2021/data-input-sources.

**Funding:** The author(s) received no specific funding for this work.

**Competing interests:** The authors have declared that no competing interests exist.

per 100,000 in 1990 to 22.4 per 100,000 in 2021, with an average annual percentage change (AAPC) of -6.64% (p < 0.001). Mortality rates rose from 14.2 per 100,000 in 1990 to 79.7 in 2000 but decreased to 13.6 by 2021, with an AAPC of -1.19%. DALYs peaked in 2010 at 1,695.1 per 100,000 and declined to 770.9 per 100,000 in 2021, with an AAPC of -1.43%. Higher incidence (28.1) and mortality (17.7) were observed in females compared to males (16.7 and 9.6). The Gambela region reported the highest burden, with an incidence rate of 176.3 per 100,000, a mortality rate of 78.4 per 100,000, and 4,220.7 DALYs per 100,000.

## Conclusion

Ethiopia accounts for 1.8% of the global HIV/AIDS prevalence. While significant progress has been made in recent years, sustained efforts are necessary to further reduce the burden of the disease. The disproportionate impact on females and the Gambela region underscores persistent disparities in prevention and care. To achieve equitable health outcomes across the country, it is crucial to implement targeted, evidence-based interventions that address these gaps in the HIV/AIDS response.

## Introduction

The HIV/AIDS epidemic has been a formidable public health challenge impacting millions of lives worldwide and exerting immense strain on healthcare systems, particularly in low- and middle-income countries [1]. Characterized by its ability to compromise the immune system and increase vulnerability to life-threatening infections and cancers, HIV/AIDS has caused widespread mortality and disability over the past four decades [2,3]. Sub-Saharan Africa has been disproportionately affected by this epidemic, accounting for a significant percentage of global HIV/AIDS cases, with Ethiopia standing among the hardest-hit countries in the region[4,5]. The burden of HIV/AIDS in Ethiopia has not only affected the health and well-being of individuals but also contributed to considerable socio-economic challenges, including reduced workforce productivity, increased healthcare costs, and widespread poverty due to loss of income from affected individuals [6,7].

Despite various global health initiatives and national campaigns aimed at reducing HIV transmission and improving access to antiretroviral therapy (ART), Ethiopia continues to grapple with high rates of HIV incidence and AIDS-related mortality [8,9]. The disease affects vulnerable populations at a greater rate, including women, young people, and marginalized groups, intensifying existing inequalities in healthcare access and economic opportunity [10]. Furthermore, the epidemic has led to significant and persistent consequences, including years of life lost (YLL), years lived with disability (YLD), and cumulative disability-adjusted life years (DALYs), which collectively represent the profound toll on individual and public health [11,12].

Tracking and understanding these impacts are essential for shaping effective health policies and intervention strategies [13]. The Global Burden of Disease (GBD) study offers a systematic and comparative framework for evaluating disease burden, providing insights into the trends and patterns of HIV/AIDS and its multifaceted impacts on population health [14]. Through metrics like YLL, YLD, and DALY, the GBD study captures not only the direct effects of HIV/AIDS but also its enduring consequences on the quality and duration of life [15]. Despite the importance of these metrics, updated evidence on the full burden of HIV/AIDS remains limited in Ethiopia.

This analysis of GBD data from 1990 to 2021 for Ethiopia focuses on the incidence, mortality, and disability metrics associated with HIV/AIDS. By examining these factors over three decades, this study aims to elucidate key trends, assess the outcomes of public health interventions, and identify areas that require enhanced focus. Such findings are critical for healthcare stakeholders, policymakers, and international organizations dedicated to reducing the burden of HIV/AIDS in Ethiopia and informing sustainable strategies for long-term epidemic control. This study also explores the regional distribution of HIV/AIDS burden within Ethiopia, mapping how incidence, mortality, and disability metrics vary across different regions. Ultimately, this research seeks to contribute valuable insights that can guide future policy-making, support resource allocation, and foster better health outcomes for individuals affected by HIV/ AIDS in Ethiopia.

## Methods

### Data source

For the study, we utilized data from the 2021 Global Burden of Disease (GBD) Study, covering the years 1990 to 2021, to examine HIV/AIDS prevalence, mortality, years of life lost (YLLs), years lived with disability (YLDs), and disability-adjusted life years (DALYs). This dataset includes rates per 100,000 population for HIV/AIDS prevalence, mortality, and DALYs, along with 95% uncertainty intervals (UIs). The data can be accessed via the Global Health Data Exchange (GHDx) at the Institute for Health Metrics and Evaluation (IHME), University of Washington (http://ghdx.healthdata.org/gbd-results-tool).

GBD is a systematic, scientific effort to quantify the comparative magnitude of health loss caused by diseases and injuries by age, sex, and location over time. GBD 2021 involves a comprehensive analysis. Data was gathered from various sources, including censuses, household surveys, vital statistics, air pollution monitoring, civil registration, disease registries, healthcare utilization, satellite imagery, disease alerts, and more. The Cause of Death Ensemble model and spatiotemporal Gaussian process regression were employed to calculate cause-specific death rates and cause fractions [16].

The study adheres to the Guidelines for Accurate and Transparent Health Estimates Reporting (GATHER)[17]. (supplementary file 1).The specific search criteria used in the "Search" interface were as follows: GBD Estimate (HIV/AIDS cases of death or disability), Measure (Incidence, Deaths), Metric (Count, Percent, Rate), Cause (HIV/AIDS), Location (Ethiopia), Age (All ages, age-standardized, and specific age groups from < 5 years to 95 + years), Sex (Both, Male, Female), and Year (1990–2021). The data included age-standardized incidence rates (ASIR), incidence counts, age-standardized disability-adjusted life year (DALYs) rates, and DALYs counts. Age-standardized death rates (ASDR) and death counts were available from 1990 onwards. Additionally, data from nine regions and two sub-cities were analyzed to map the distribution of the HIV/AIDS burden across the country. Age-standardized disability-adjusted life years (DALYs) is a measure used to assess the overall burden of disease in a population, accounting for both the years of life lost due to premature death (YLL) and the years lived with disability (YLD). The age-standardization process adjusts the DALYs value to a standard age distribution, allowing for comparisons between populations with different age structures or over time.

### Statistical analyses

The burden of HIV in the country was quantified using incidence, mortality, years of life lost, years lived with disability, and disability-adjusted life years. Age-standardized rates (ASRs)

for specific age groups, along with estimated values and 95% uncertainty intervals (UIs), were extracted from GBD 2021. ASR was computed using the formula:

$$ASR = \frac{\sum_{i=1}^{N} a_i w_i}{\sum_{i=1}^{N} wi} c = \text{Where } a_i \text{ represents the age-specific rate for the } ith \text{ age group and}$$

$w_i$ denotes the number of individuals (or the weight) in the same age group within the GBD 2021 standard population. N is the number of age groups.

We calculated the percentage change in the number of incident cases and deaths by comparing the case counts from 1990 to 2021.

$$\text{Percent change} = \frac{(y_{1990} - y_{2021}) \times 100}{y_{2021}} = \text{Y can refer to either the total number of incident}$$

cases or the total number of deaths.

We employed Jointpoint regression to analyze the trends in the incidence and mortality rates of HIV/AIDS in Ethiopia from 1990 to 2021. This method allows us to identify points in time where the rate of change in the data significantly shifts. The regression model fits multiple linear segments to the data, with each segment representing a distinct trend. Joinpoints, where the slope of the trend changes, were identified to examine periods of acceleration or deceleration in HIV/AIDS burden. We calculated the Annual Percentage Change (APC), which reflects the rate of change in the data within that segment. The Average Annual Percentage Change (AAPC) was then computed by averaging the APC values across all identified segments, weighted by the duration of each segment. The AAPC provides a summary measure of the overall rate of change in HIV/AIDS incidence and mortality over the study period, accounting for any shifts in the trends. Significant of change was determined at P-value of 0.05.

We utilized ArcGIS Pro to visualize and map the distribution of the HIV burden across Ethiopia. Geographic data from 9 regions and two sub-cities were analyzed to create detailed maps representing the incidence and mortality rates of HIV. Using spatial analysis tools in ArcGIS Pro, we visualized the geographic variation in the HIV burden, enabling a clear depiction of areas with higher or lower rates

### Ethical approval

This study did not require ethical board approval since it relied on a publicly available dataset and did not involve human or animal trials.

## Results

### Trends of HIV/AIDS prevalence in Ethiopia (1990-2021)

In 2021, a total of 715,839 people (95% UI: 568,889.58–914,511.59) were living with HIV in Ethiopia. From 1990 to 2021, HIV/AIDS prevalence in Ethiopia showed a notable rise in the early 1990s, peaking in 1998 across all sex groups. The overall prevalence increased from 0.3979% in 1990 to 1.1715% in 1998, after which it gradually declined to 0.6722% by 2021. Males followed a similar trend, with rates rising from 0.2778% in 1990 to 0.8313% in 1998, before decreasing to 0.4819% in 2021. Females consistently had higher prevalence rates throughout the period, beginning at 0.5186% in 1990, peaking at 1.5189% in 1998, and declining to 0.8637% by 2021. (Fig 1).

### Trends in mortality rate per 100 K attributable to HIV/AIDS in Ethiopia (1990-2021)

From 1990 to 2021, the overall death rate per 100,000 population saw a modest decline, decreasing from 14.2 to 13.6. In the < 20 years age group, the death rate dropped from 4.4

### Trends of HIV/AIDS in Ethiopia (1990-2021)

**Fig 1. Trends of HIV/AIDS in Ethiopia (1990-2021).**

in 1990 to 3.3 in 2021. For the 20-54 years age group, the rate decreased more significantly from 29.8 in 1990 to 24.1 in 2021, with a notable mid-point drop to 142.9 in 2005. The > 54 years age group showed a different trend, with the rate increasing from 16.9 in 1990 to 48.8 in 2021, though a mid-point in 2005 saw a lower rate of 53.1, indicating fluctuations rather than a steady decline. When comparing by sex, females consistently had higher death rates than males. In 1990, the death rate for males was 9.8, which slightly decreased to 9.6 by 2021, with a mid-point in 2005 showing a rate of 45.3. For females, the rate dropped from 18.6 in 1990 to 17.7 in 2021, with a mid-point rate of 84.0 in 2005, suggesting a more substantial overall reduction. Overall, while all groups experienced reductions, the decline was more pronounced in the 20-54 years age group, and the most significant fluctuations were observed in those aged > 54 years, particularly among females. (Table 1).

### Disability-adjusted life years

From 1990 to 2021, HIV Disability-Adjusted Life Years (DALYs) decreased from 4,461,606 to 770,900, with an average annual percentage change (AAPC) of -17.92%. DALYs for females dropped from 2,937,811 to 541,421 (AAPC: -10.36%), and for males from 1,523,794 to 239,879 (AAPC: -11.44%). In the < 20 years age group, DALYs showed minimal change (AAPC: 0.46%). The 20-54 years group saw a substantial decrease (AAPC: -13.28%). In contrast, the > 54 years age group experienced an increase in DALYs (AAPC: 3.43%). (Table 2).

### Years lived with disability and years of life lost

In 2021, the YLD due to HIV/AIDS reached 78,772.2, a significant rise from 28,674.2 in 1990, with the most substantial increase seen in individuals over 54 years, whose YLD surged from 1,351.3 in 1990 to 10,860.8 in 2021, reflecting an AAPC of 6.22%. For females, YLD amounted to 50,557.4, up from 18,578.9, while for males, it reached 28,214.7, increasing from 10,095.3;

**Table 1. Trends in Mortality rate per 100 K Attributable to HIV/AIDS in Ethiopia (1990-2021).**

| Year | Sex | | Age in years | | | Over all |
|---|---|---|---|---|---|---|
| | Male (95 CI) | Female (95 CI) | <20 years | 20-54 years | >54 years | |
| 1990–1994 | 116.6 (66.1–200.0) | 213.1 (135.7 - 351.8) | 48.5 (27.1–78.7) | 355.0 (217.0–609.5) | 218.3 (134.9–347.3) | 192.6 (117.6–315.4) |
| 1995–1999 | 269.8 (161.4–427.2) | 309.2 (309.2 - 309.2) | 82.3 (49.3–127.3) | 829.4 (512.7–1293.4) | 571.8 (406.2–779.1) | 353.2 (219.2–550.2) |
| 2000–2004 | 226.8 (150.4–329.1) | 286.6 (286.6 - 286.6) | 70.9 (44.5–104.1) | 706.5 (474.8–1024.2) | 471.6 (362.4–593.4) | 312.4 (215.9–439.5) |
| 2005–2009 | 103.3 (76.4–147.0) | 146.9 (146.9 - 146.9) | 47.8 (31.2–69.0) | 292.1 (220.3–406.9) | 201.6 (135.1–287.6) | 146.8 (112.0–201.8) |
| 2010–2014 | 66.9 (48.0–96.7) | 96.5 (96.5 - 96.5) | 32.1 (20.1–47.0) | 188.6 (141.3–260.8) | 152.6 (98.5–222.5) | 99.1 (74.6–135.0) |
| 2015–2019 | 100.1 (65.9–147.0) | 215.5 (147.2 - 317.8) | 46.2 (27.4–71.4) | 400.1 (259.0–607.9) | 303.8 (209.7–422.4) | 188.9 (123.6–282.9) |
| 2020–2021 | 71.6 (40.7–120.1) | 100.2 (65.4 - 156.8) | 14.4 (8.1–22.9) | 106.8 (69.2–173.7) | 80.6 (49.9–124.3) | 52.2 (34.0–82.6) |

**Table 2. Disability-Adjusted Life Years of HIV in Ethiopia from 1990-2021.**

| Categories | 1990 (95UI) | 2000 (95UI) | 2010 (95UI) | 2021 (95UI) | AAPC (95UI) 1990-2021 |
|---|---|---|---|---|---|
| Number of DALYs | | | | | |
| Over all | 446106.6 (268183.4 -763760.1) | 3114570.7 (1940937.2-4747749.8) | 1443676.9 (1077386.8-1987034.3) | 770.9 (569.6-1059.8) | -17.92 (-51.97- 39.75) |
| Sex | | | | | |
| Female | 293371.8 (183193.8-494225.1) | 1999747.6 (1267636.3-3033463.9) | 929841.7 (701265.2-1274512.4) | 541421.4 (405935.1-732142.5) | 0.96 (-10.86-14.07) |
| Male | 152734.8 (84340.5-272612.0) | 1114823.1 (664825.1-1731926.0) | 513835.2 (376331.3-723084.2) | 298397.9 (214724.0-430896.8) | 1.11 (-11.05-14.66) |
| Age | | | | | |
| <20 years | 112469.6 (58761.2-198025.6) | 617774.0 (370149.9-935118.0) | 398273.7 (265835.4-566928.3) | 158049.1 (93334.5-246484.0 | 0.46 (-13.38-16.41) |
| 20-54 years | 316954.7 (192747.4-554430.1) | 2355160.8 (1459712.6-3659099.8) | 984897.6 (740026.6-1370762.7) | 609942.8 (460736.7-825600.8) | 0.98 (-10.78-14.07) |
| >54 years | 16682.3 (9538.1-28203.9) | 141635.9 (104286.5-186206.9) | 60505.6 (40015.3-86409.9) | 71827.5 (48561.2-99621.2) | 3.43 (-5.82-13.37) |
| DALYs per 100 000 | | | | | |
| Over all | 882.2 (530.3-1510.3) | 4552.2(2836.9- 6939.3) | 1695.1(1265.1-2333.2) | 770.9 (569.6 - 1059.8) | -1.43(-12.73 -11.06) |
| Sex | | | | | |
| Female | 1162.9(726.1- 1959.0) | 5907.8 (3745.0 - 8961.7) | 2205.1 (1663.0 - 3022.4) | 1002.0 (751.3 - 1355.0) | -1.47(-12.69-10.94) |
| Male | 602.7 (332.8 - 1075.8) | 3224.9 (1923.2 - 5010.0) | 1195.0 (875.3 - 1681.7) | 543.5 (391.1 - 784.8) | -1.3(-12.8-11.36) |
| Age | | | | | |
| <20 years | **381.7 (199.4 - 672.0)** | **1526.9 (914.8 - 2311.2)** | **809.9 (540.6 - 1152.9)** | **276.3 (163.2 - 430.9)** | -1.64(-14.26-12.75) |
| 20-54 years | 1771.8(1077.5 - 3099.4) | 9803.6 (6076.2 - 15231.4) | 3170.8 (2382.5 - 4413.1) | 1358.6(1026.3 - 1839.0) | -1.94(-13.62-11.11) |
| >54 years | **519.2 (296.9 - 877.8)** | **3599.6 (2650.4 - 4732.4)** | **1227.4 (811.7 - 1752.9)** | **1049.3 (709.4 - 1455.3)** | 0.95(-8.63-11.13) |

their respective AAPCs were 2.66% and 2.72%, indicating similar annual growth rates for both sexes. The overall AAPC for YLD from 1990 to 2021 was 2.68%. In the same year, YLL due to HIV/AIDS was 761,047.1, an increase from 417,432.5 in 1990. Females experienced YLL of 490,864.0, up from 274,792.9, and males had YLL of 270,183.2, rising from 142,639.5. The YLL for those under 20 years increased from 108,844.7 in 1990 to 150,303.6 in 2021, while those aged 20-54 years saw a substantial rise from 293,256.7 to 549,776.9, and the > 54 years group experienced growth from 15,331.0 to 60,966.6. The overall AAPC for YLL from 1990 to 2021 was 0.89%. (Table 3).

**Table 3. Years of Life Lost and Years Lived with Disability of HIV in Ethiopia from 1990-2021.**

| Categories | 1990 (95UI) | 2000 (95UI) | 2010 (95UI) | 2021 (95UI) | AAPC (95UI) 1990-2021 |
|---|---|---|---|---|---|
| *Years Lived with Disability* | | | | | |
| Over all | 28,674.2 (13,122.3 - 46,869.9) | 111,385.8 (72,893.4 - 154,922.9) | 86,789.8 (57,263.5 -120,949.1) | 78,772.2 (49,210.5 - 118,982.9) | 2.68 (-4.71-10.49) |
| Sex | | | | | |
| Female | 18,578.9 (8,606.3 - 30,553.2) | 70,397.0 (45,290.5 - 97,962.8) | 55,244.2 (36,348.6 - 77,816.3) | 50,557.4 (31,530.9 - 76,980.4) | 2.66 (-4.56-10.27) |
| Male | 10,095.3 (4,588.3 - 16,240.7) | 40,988.8 (26,632.7 - 58,634.2) | 31,545.6 (20,242.3 - 45,402.6) | 28,214.7 (17,185.0 - 43,452.3) | 2.72 (-4.98-10.88) |
| Age | | | | | |
| <20 years | 3,624.9 (1,643.7 - 6,030.6) | 14,311.2 (8,860.0 - 20,793.2) | 12,724.3 (8,352.8 - 18,094.8) | 7,745.4 (4,667.4 - 11,826.9) | 2.04 (-8.38-13.55) |
| 20-54 years | 23,698.0 (10,804.1 - 38,795.4) | 90,999.5 (59,513.0 - 126,374.8) | 68,031.0 (44,794.9 - 97,592.1) | 60,165.9 (37,900.2 - 93,817.6) | 2.41 (-5.01-10.23) |
| >54 years | 1,351.3 (599.1 - 2,321.2) | 6,075.2 (3,446.3 - 10,755.8) | 6,034.6 (3,536.1 - 9,758.6) | 10,860.8 (7,010.3 - 15,833.2) | 6.22 (1.49-11.15) |
| Categories | | | | | |
| *Years of Life Lost* | | | | | |
| Over all | 417,432.5 (254,422.7 724,371.1) | 3,003,184.9 (1,851,662.6 - 4,629,832.3) | 1,356,887.1 (1,000,265.1 - 1,904,124.3) | 761,047.1 (560,300.4 -1,057,675.0) | 0.89 (-11.36-14.55) |
| Sex | | | | | |
| Female | 274,792.9 (173,376.0 -466,836.6) | 1,929,350.6 (1,220,226.3 - 2,954,752.3) | 874,597.4 (652,156.0 - 1,216,614.9) | 490,864.0 (367,961.6 - 671,244.8) | 0.83 (-11.29-14.34) |
| Male | 42,639.5 (79,064.9 - 261,146.5) | 1,073,834.2 (633,467.6 - 1,680,356.2) | 482,289.6 (351,655.0 - 690,511.1) | 270,183.2 (191,461.4 - 392,780.8) | 0.99 (-11.48-14.94) |
| Age | | | | | |
| <20 years | 108,844.7 (56,616.8 - 192,931.2) | 603,462.8 (360,361.6 - 919,116.0) | 385,549.5 (255,119.1 - 553,700.9) | 150,303.6 (88,358.3 - 234,408.4) | 0.39 (-13.57-16.51) |
| 20-54 years | 293,256.7 (179,773.1 - 522,630.2) | 2,264,161.4 (1,383,481.7 - 3,562,821.6) | 916,866.6 (683,825.7 - 1,300,765.7) | 549,776.9 (412,935.1 - 755,286.9) | 0.8671 (-11.19-14.33) |
| >54 years | 15,331.0 (8,723.2 - 26,230.6) | 135,560.7 (98,602.5 - 180,132.6) | 54,471.0 (35,504.2 - 77,813.0) | 60,966.6 (39,077.7 - 89,213.1) | 3.12 (-6.57-13.51) |

## Age-standardized incidence, mortality, and disability-adjusted life years (DALYs) of HIV (1990–2021)

The table shows a significant decline in age-standardized HIV incidence from 1990 (175.4 per 100,000) to 2021 (22.4 per 100,000), with an average annual percentage change (AAPC) of -6.64% (p < 0.001). Mortality rates increased from 14.2 per 100,000 in 1990 to 79.7 in 2000 but declined to 13.6 by 2021, with an AAPC of -1.19%, but the change is not significant (p > 0.05). Disability-adjusted life years (DALYs) peaked in 2010 at 1,695.1 per 100,000 before decreasing to 770.9 in 2021, with an AAPC of -1.43%, but the change is not significant (p > 0.05). In 2021, a 839,819.31 (95UI: 620,558.56 - 1,154,483.92) DALYs was observed. (Table 4).

The incidence of HIV/AIDS saw a sharp rise from 1990 to 1992, with an APC of 16.85%, followed by stabilization between 1992 and 1995 (APC: -0.27%). From 1995 onwards, a steady and significant decline occurred, with the steepest drop between 1995 and 2004 (APC: -9.95%). This downward trend continued from 2004 to 2010 (APC: -5.84%), with further declines between 2010 and 2014 (APC: -9.06%), 2014 to 2018 (APC: -6.97%), and 2018 to 2021 (APC: -10.00%). (Fig 2).

**Table 4. Age-standardized incidence, mortality, and disability-adjusted life years (DALYs) of HIV (1990–2021).**

| Categories | 1990 (95UI) | 2000 (95UI) | 2010 (95UI) | 2021 (95UI) | AAPC (95UI) 1990-2021 | P-value |
|---|---|---|---|---|---|---|
| Age-standardized incidence rate | | | | | | |
| Over all | 175.4(71.3-299.4) | 126.7 (86.2-186.4) | 57.7(42.5-77.9) | 22.4 (12.9-45.8) | -6.64 (-9.46 - 3.64) | <0.001 |
| Sex | | | | | | |
| Female | 212.6 (88.9 – 374.4) | 154.5 (103.8 – 226.3) | 70.5 (51.1 – 96.3) | 28.1 (16.1 – 59.3) | -6.46(-9.13 - 3.70) | <0.001 |
| Male | 136.2 (55.5 – 244.6) | 98.6 (66.1 – 148.8) | 44.7(32.4 – 62.3) | 16.7 (9.6 – 32.6) | -6.72(-9.86 - 3.56) | <0.001 |
| Age-standardized mortality rate | | | | | | |
| Over all | 14.2 (8.7 – 24.4) | 79.7 (50.0 – 121.5) | 28.4 (21.8 – 39.1) | 13.6 (10.3 – 18.3) | -1.19 (-12.59-11.47) | >0.05 |
| Sex | | | | | | |
| Female | 18.6 (11.9 – 31.3) | 102.3 (66.2 – 154.2) | 36.9 (28.4 – 50.3) | 17.7 (13.6 – 23.7) | -1.20(-12.49-11.35) | >0.05 |
| Male | 9.8 (5.4 – 18.0) | 57.6 (34.6 – 90.2) | 20.1 (14.8 – 28.8) | 9.6 (6.9 – 13.9) | -1.17(-12.78- 11.77) | >0.05 |
| Disability-adjusted life years | | | | | | |
| Over all | 552.2 (2836.9 – 6939.3) | 882.2 (530.3 – 1510.3) | 1695.1 (1265.1 – 2333.2) | 770.9 (569.6 – 1059.8) | -1.43 (-12.73-11.05) | >0.05 |
| Sex | | | | | | |
| Female | 602.7 (332.8 – 1075.8) | 3224.9 (1923.2 – 5009.9) | 1195.0 (875.3 – 1681.7) | 543.5 (391.1 – 784.8) | -1.47 (-12.69 - 10.93) | >0.05 |
| Male | 3224.88 (1923.16 – 5009.99) | 1195.05 (875.25 – 1681.71) | 602.74 (332.83 – 1075.81) | 543.47 (391.08 – 784.79) | -1.34 (-12.81 - 11.36) | >0.05 |

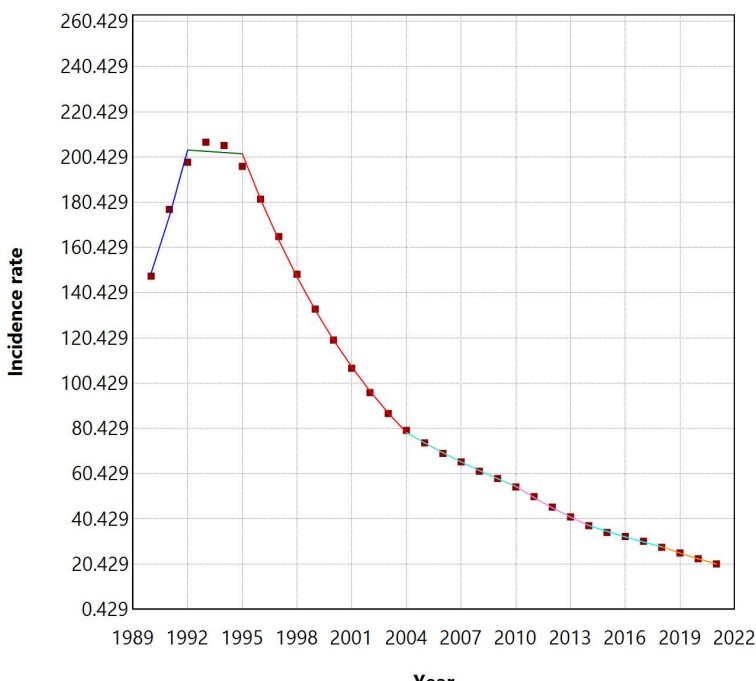

**Fig 2. Age-standardized incidence of HIV/AIDS (1990–2021).**

The mortality rate attributable to HIV/AIDS increased sharply from 1990 to 1993 (APC: 38.39%) and continued to rise at a slower pace between 1993 and 1996 (APC: 18.97%). From 1996 to 2000, the increase slowed further (APC: 6.07%) before peaking in 2000. Following this peak, the mortality rate declined gradually from 2000 to 2004 (APC: -2.33%), with a significant decrease from 2004 to 2010 (APC: -15.53%), and continued to fall from 2010 to 2021 (APC: -5.87%). (Fig 3).

From 1990 to 1996, HIV/AIDS DALYs increased significantly, with an APC of 37.03% and 17.69% respectively, peaking in 2000. Following this, the trend reversed, with a gradual decline in DALYs from 2000 to 2004 (APC: -2.97%) and a sharper decrease from 2004 to 2010 (APC: -14.41%). This decline continued through 2021, though at a slower rate (APC: -6.28%). (Fig 4).

## Distribution of burden of HIV/AIDS across regions of Ethiopia

An analysis of the geographical distribution of the HIV/AIDS burden shows that the highest age-standardized incidence rate is observed in the Gambella region, with 176.3 (95UI:65.7-360.5) cases per 100,000 population, followed by Addis Ababa (the capital city of Ethiopia) with 78.4 (95UI: 30.3-169.1) cases per 100,000 population and Dire Dawa with 58.4 (95UI:17.6-132.4) cases per 100,000 population. (Fig 5).

The highest age-standardized mortality rate associated with HIV/AIDS is observed in the Gambella region, with 78.4 (95UI: 46.01- 140.4) per 100,000 population, whereas the lowest mortality rate, 10.6 (95UI: 5.9- 20.4) per 100,000 population, is observed in the SNNPR. (Fig 6).

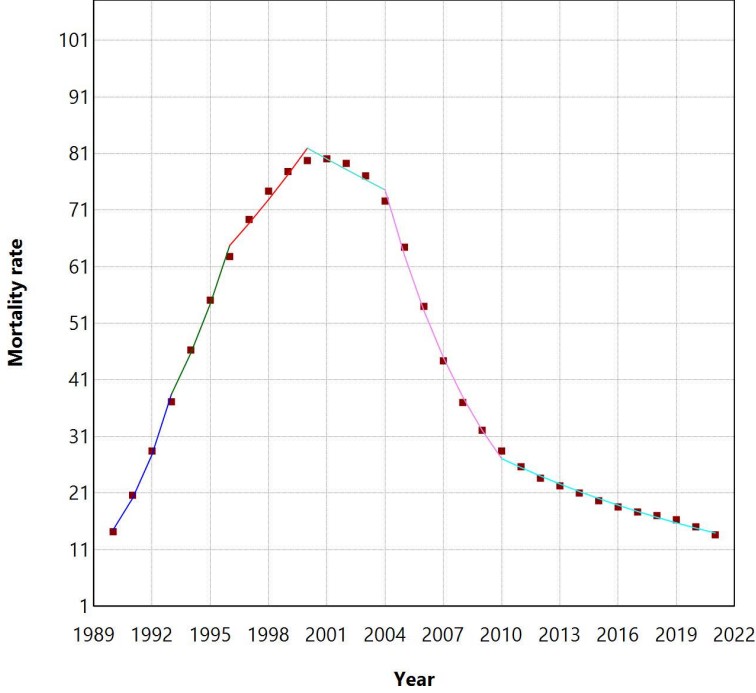

**Fig 3. Age-Standardized Mortality Rate Attributable to HIV/AIDS (1990-2021).**

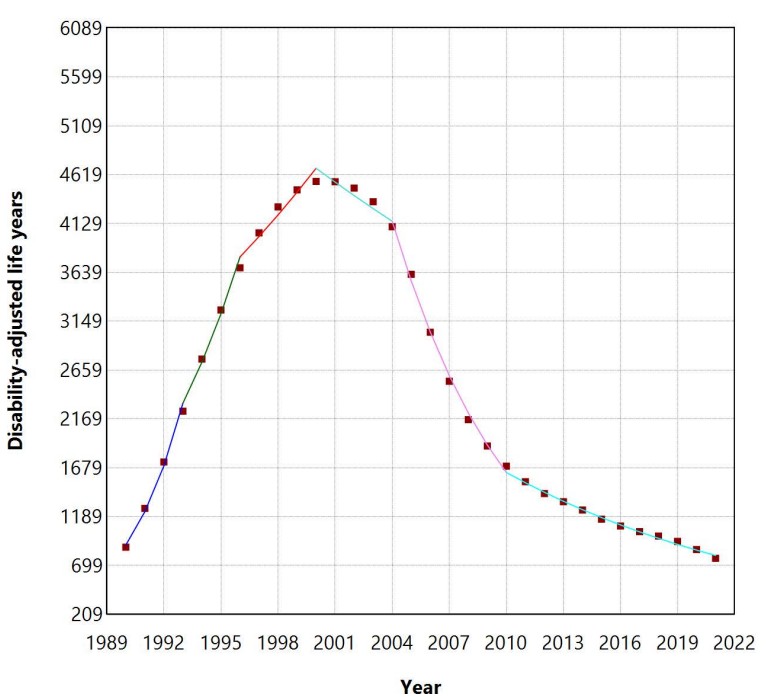
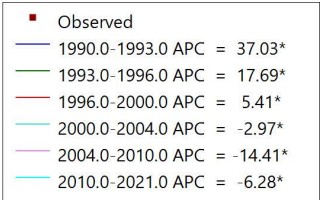

**Fig 4. Age-Standardized Disability-Adjusted Life Years (DALYs) due to HIV/AIDS (1990–2021).**

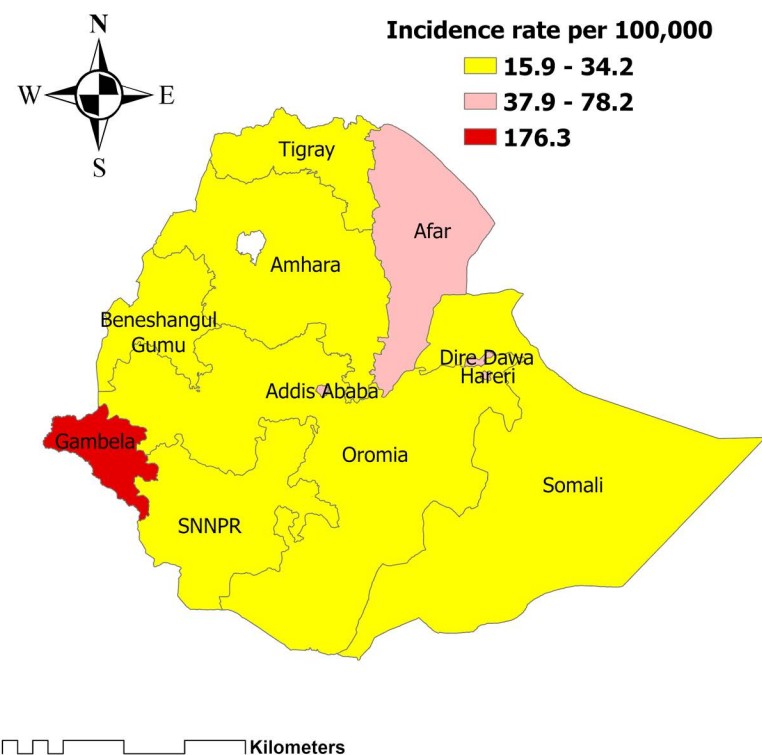

**Fig 5. Distribution of age-standardized incidence rate of HIV/AIDS across regions of Ethiopia.**

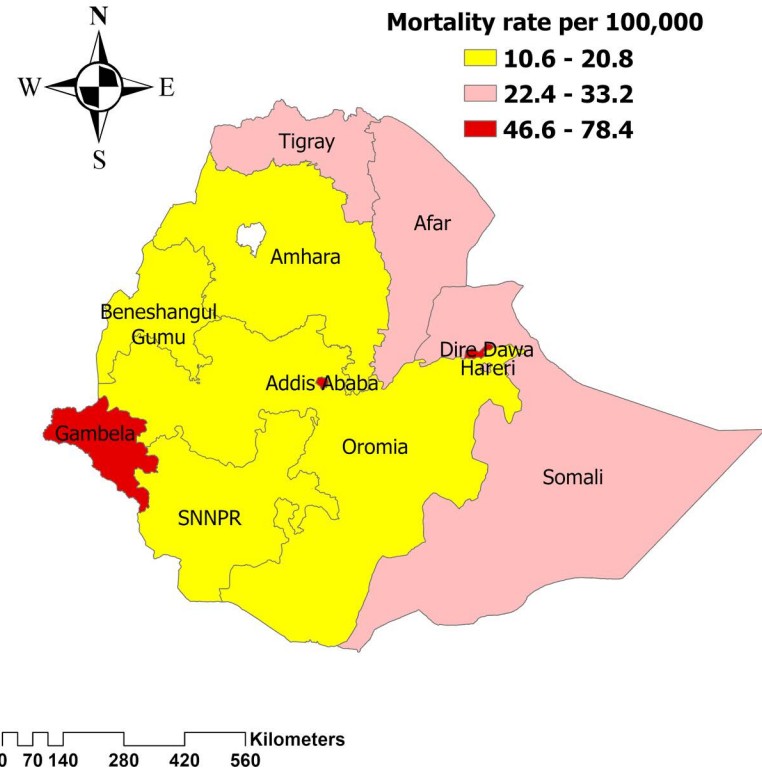

**Fig 6. Distribution of age-standardized mortality rate associated with HIV/AIDS across regions of Ethiopia.**

The highest age-standardized DALYs were also found in the Gambella region, with 4,220.7 (95UI: 2,515.2 - 7,592.7), followed by Addis Ababa with 3,271.3 (95UI: 2,180.2 - 5,339.9) Dire Dawa with 2,465.7 (95UI: 1,209.6 - 4,170.34) - and Afar with 1,701.7 (95UI: 974.3- 2,952.9). (Fig 7).

## Discussion

This study aimed to assess the overall burden of HIV/AIDS in Ethiopia using data from the Global Burden of Disease Study. In 2021, a total of 715,839 people were living with HIV in Ethiopia, representing a prevalence of 0.67%. This figure accounts for 1.8% of the global HIV/AIDS prevalence, with an estimated more than 40 million people living with HIV/AIDS worldwide [18]. The proportion attributed to Ethiopia demonstrates that, while the country is not among the highest contributors to the global burden compared to other Sub-Saharan African countries, it remains a critical area for intervention, particularly in Sub-Saharan Africa, where HIV/AIDS continues to be a leading cause of morbidity and mortality.

The overall prevalence increased from 0.3979% in 1990 to 1.1715% in 1998, after which it gradually declined to 0.6722% by 2021. The introduction of global and national HIV/AIDS programs likely played a critical role in controlling the epidemic and reducing the number of new infections.

The reduction of the HIV/AIDS burden can be attributed to the implementation of a variety of targeted programs. These include school-based HIV education programs, condom distribution initiatives, pre-exposure prophylaxis (PrEP), integrated Voluntary Medical Male Circumcision (VMMC) services, targeted outreach efforts, peer service provider (PSP) programs, and specialized HIV services for prisoners. Despite the decline, the prevalence

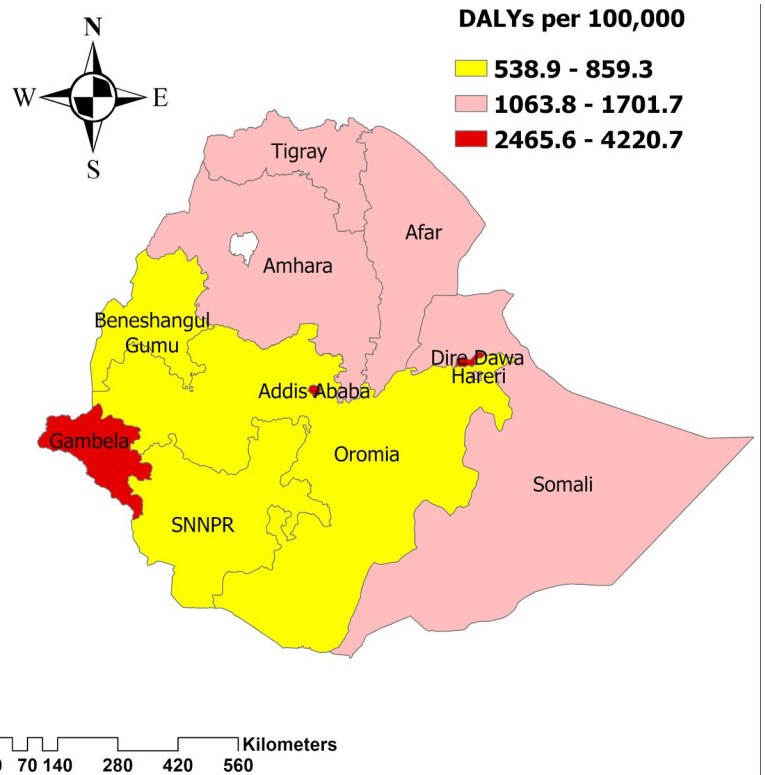

**Fig 7. Distribution of age-standardized DALYs of HIV/ADIS across regions of Ethiopia.**

remains a significant public health challenge, with substantial numbers of individuals requiring ongoing treatment and care. This trend underscores the need for sustained investment in prevention, treatment, and education programs to maintain the gains achieved and further reduce the burden of HIV/AIDS in Ethiopia.

Despite a decrease in prevalence and incidence, females still disproportionately suffer from the burden of HIV. In 2021, significant inequalities were observed in prevalence, with males at 0.48% compared to females at 0.86%. The gender disparity in HIV prevalence can be attributed to multiple factors that increase women's susceptibility to the virus. These include starting sexual activity at a younger age, engaging in sexual transactions with older men, and facing sexual violence [19]. Additionally, biological factors contribute to a higher likelihood of HIV transmission from men to women, particularly among younger women [20].

Between 1990 and 2021, the overall mortality rate per 100,000 population demonstrated a gradual decline, decreasing from 14.2 to 13.6. A recent systematic review conducted in Ethiopia also revealed that 5% to 40.8% of patients died during the follow-up period [21]. While this modest reduction indicates some progress in public health and healthcare delivery, it also highlights the slow pace of improvement and the need for more targeted and effective policy measures to accelerate health outcomes. The slight decline in mortality highlights the need for policies that improve healthcare access and quality, especially in underserved areas. Investments in healthcare infrastructure, preventive care, and workforce training are essential. Strengthening health data systems and community-based programs can support evidence-based planning and enhance health outcomes.

In 2021, a significant burden of disease was observed, with 839,819.31 Disability-Adjusted Life Years (DALYs) attributed to HIV/AIDS. This marks notable progress from 2016, when

1.1 million DALYs were reported [6]. The decline in DALYs indicates improvements in HIV prevention, treatment, and care strategies over the past few years. However, the continuing substantial burden emphasizes the need for sustained efforts to further reduce the impact of HIV/AIDS, particularly through expanded access to antiretroviral therapy, enhanced prevention programs, and addressing social determinants that contribute to the spread of the virus.

Despite significant progress over the past 30 years, the COVID-19 pandemic may present challenges to the ongoing efforts to control the HIV epidemic in Ethiopia. A recent study highlighted a notable decline in several key indicators, including the number of HIV tests, consultations, viral load tests, antiretroviral therapy (ART) initiations, and ART adherence [22]. Similar studies also assert that the pandemic has disrupted routine clinical care and treatment, leading to treatment failures and increased drug resistance [23,24].

The findings indicate that the Gambela region (western Ethiopia) faces a disproportionately high burden of HIV/AIDS, as reflected in higher incidence, mortality, and DALYs compared to other regions. Addis Ababa and Dire Dawa also showed significant HIV/AIDS prevalence, but the extent of the burden in Gambela is notably greater. A recent study corroborates these findings, highlighting a persistent and high prevalence of HIV/AIDS in these regions [25,26,27]. There is a need for region-specific policies that focus on improving access to HIV prevention, testing, and treatment services, particularly in the Gambela region. This includes enhancing healthcare infrastructure, increasing availability of antiretroviral therapy (ART), and expanding HIV awareness programs. Additionally, addressing social determinants such as poverty, education, and healthcare access can help reduce vulnerability to HIV. Ensuring equitable distribution of resources and strengthening local healthcare systems are critical steps to improving outcomes in these high-burden regions.

The study has several strengths and a few limitations. Several strengths include providing valuable insights over a long period (1990–2021) using data from the Global Burden of Disease Study, ensuring credibility and national relevance. It also offers a comprehensive national perspective on the HIV/AIDS burden, making it useful for policymakers and healthcare planners. Additionally, the use of a globally recognized dataset enhances the reliability of the findings. However, the study is limited by its reliance on secondary data, which may not capture local nuances or emerging trends. The lack of primary data means specific regional challenges might be overlooked, and variations in reporting methods over time could affect data consistency.

## Conclusion

The findings indicate significant progress in the fight against HIV/AIDS in Ethiopia, demonstrated by notable declines in incidence, mortality, and DALYs over the past three decades. Despite these achievements, challenges such as persistent gender disparities and regional inequalities remain, with the Gambela region continuing to experience a high burden of the disease. While past interventions have contributed to improvements, sustained efforts are necessary to maintain these gains and address ongoing disparities.

### Recommendations and targeted interventions

Policymakers should focus on gender-sensitive, age-specific interventions. For women, empowerment programs, access to reproductive health services, and prevention methods like PrEP should be prioritized. For adolescent girls and young women (AGYW), comprehensive sexual education, skills training, and economic empowerment programs are essential to reduce vulnerability. For older adults, HIV awareness and treatment services should be targeted to ensure access, including tailored counseling and ART.

In high-burden regions like Gambela, region-specific interventions are needed. These should include improving access to care, enhancing community-based prevention, and reducing stigma through outreach and education.

Prevention efforts should address high-risk populations with targeted programs for those engaging in risky behaviors, such as sex workers and drug users, with age-specific harm-reduction programs.

To ensure equitable access to treatment, focus on women of reproductive age by providing ART and maternal health services to prevent mother-to-child transmission. For children and adolescents, expanding access to pediatric ART and involving caregivers in treatment plans will be vital.

Maintaining strong surveillance systems that capture age- and gender-specific data will guide targeted interventions. Strengthening community-level health programs by engaging local leaders and establishing peer support networks for different age groups is essential.

## Supporting information

**supplementary file 1: GATHER checklist.**
(PDF)

## Author contributions

**Conceptualization:** Tegene Atamenta Kitaw, Molla Azmeraw, Befkad Derese Tilahun, Alemu Birara Zemariam, Addisu Getie, Molalign Aligaz Adisu, Ribka Nigatu Haile.

**Data curation:** Tegene Atamenta Kitaw, Befkad Derese Tilahun, Alemu Birara Zemariam, Addis Wondmagegn Alamaw, Abebe Merchaw Faris, Addisu Getie, Molalign Aligaz Adisu, Ribka Nigatu Haile.

**Formal analysis:** Tegene Atamenta Kitaw, Molla Azmeraw, Alemu Birara Zemariam, Abebe Merchaw Faris, Addisu Getie, Molalign Aligaz Adisu, Ribka Nigatu Haile.

**Funding acquisition:** Ribka Nigatu Haile.

**Investigation:** Tegene Atamenta Kitaw, Befkad Derese Tilahun.

**Methodology:** Tegene Atamenta Kitaw, Ribka Nigatu Haile.

**Project administration:** Tegene Atamenta Kitaw.

**Resources:** Tegene Atamenta Kitaw.

**Software:** Tegene Atamenta Kitaw, Molla Azmeraw, Biruk Beletew Abate, Addis Wondmagegn Alamaw, Abebe Merchaw Faris, Tesfaye Engdaw Habtie, Melesse Abiye Munie, Ribka Nigatu Haile.

**Supervision:** Tegene Atamenta Kitaw, Alemu Birara Zemariam, Tesfaye Engdaw Habtie, Ribka Nigatu Haile.

**Validation:** Tegene Atamenta Kitaw, Biruk Beletew Abate, Addis Wondmagegn Alamaw, Tesfaye Engdaw Habtie, Melesse Abiye Munie, Ribka Nigatu Haile.

**Visualization:** Tegene Atamenta Kitaw, Biruk Beletew Abate, Melesse Abiye Munie, Ribka Nigatu Haile.

**Writing – original draft:** Tegene Atamenta Kitaw, Molla Azmeraw, Biruk Beletew Abate, Befkad Derese Tilahun, Alemu Birara Zemariam, Addis Wondmagegn Alamaw, Abebe Merchaw Faris, Addisu Getie, Molalign Aligaz Adisu, Tesfaye Engdaw Habtie, Melesse Abiye Munie, Ribka Nigatu Haile.

**Writing – review & editing:** Tegene Atamenta Kitaw, Molla Azmeraw, Biruk Beletew Abate, Befkad Derese Tilahun, Alemu Birara Zemariam, Addis Wondmagegn Alamaw, Abebe Merchaw Faris, Addisu Getie, Molalign Aligaz Adisu, Tesfaye Engdaw Habtie, Melesse Abiye Munie, Ribka Nigatu Haile.

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
