## [Decision Letter · Decision Letter 0]

6 Jan 2025

PONE-D-24-52871The Burden of HIV/AIDS in Ethiopia: Unveiling 30 Years of Trends in Incidence, Mortality, and Disability—Insights from the Global Burden of Disease Study (1990–2021).PLOS ONE

Dear Dr. Kitaw,

Thank you for submitting your manuscript to PLOS ONE. After careful consideration, we feel that it has merit but does not fully meet PLOS ONE’s publication criteria as it currently stands. Therefore, we invite you to submit a revised version of the manuscript that addresses the points raised during the review process.==============================

We look forward to receiving your revised manuscript.

Kind regards,

Deepak Dhamnetiya, MD

Academic Editor

PLOS ONE

Journal requirements: When submitting your revision, we need you to address these additional requirements. 1. Please ensure that your manuscript meets PLOS ONE's style requirements, including those for file naming. The PLOS ONE style templates can be found at https://journals.plos.org/plosone/s/file?id=wjVg/PLOSOne_formatting_sample_main_body.pdf and https://journals.plos.org/plosone/s/file?id=ba62/PLOSOne_formatting_sample_title_authors_affiliations.pdf. 2. Please amend either the title on the online submission form (via Edit Submission) or the title in the manuscript so that they are identical. 3. Your abstract cannot contain citations. Please only include citations in the body text of the manuscript, and ensure that they remain in ascending numerical order on first mention. 4. Please include a caption for figure 2, 3 and 4. 

Reviewers' comments:

Reviewer's Responses to Questions

**Comments to the Author**

1. Is the manuscript technically sound, and do the data support the conclusions?

Reviewer #1: Yes

Reviewer #2: Yes

2. Has the statistical analysis been performed appropriately and rigorously? 

Reviewer #1: Yes

Reviewer #2: Yes

3. Have the authors made all data underlying the findings in their manuscript fully available?

Reviewer #1: Yes

Reviewer #2: Yes

4. Is the manuscript presented in an intelligible fashion and written in standard English?

Reviewer #1: Yes

Reviewer #2: Yes

5. Review Comments to the Author

Reviewer #1: 1. The Table 1 can be made more concise and data can be grouped in 5ys and shown

2. Recommendations and targeted interventions need to be specified more as to what targeted interventions can be done age wise or gender wise.

Reviewer #2: The manuscript appears to be technically sound. The results regarding the trends in HIV/AIDS in Ethiopia are supported by data from the Global Burden of Disease Study, which was analyzed over a significant time span. The methodology seems appropriate, and the findings are strengthened by the use of gender, regional, and age-standardized data. The conclusions are well-supported by the data, and the manuscript offers useful suggestions for addressing persistent HIV/AIDS-related challenges in Ethiopia.

The statistical analysis appears to be appropriate and rigorous, with sound methods used for analyzing the trends.

The manuscript is generally clear and well-structured, with a logical flow from introduction to methods, results, and conclusion. The language is mostly correct and appropriate for a scientific manuscript. However, it might benefit from further elaboration on terms like "disability-adjusted life years (DALYs)" to ensure clarity, especially for readers who may not be familiar with the concept. Additionally, I suggest standardizing the use of abbreviations throughout the manuscript for consistency and readability, particularly for terms like DALYs, GBD, ART, and others. This will enhance the clarity and uniformity of the manuscript.

6. PLOS authors have the option to publish the peer review history of their article (what does this mean? ). If published, this will include your full peer review and any attached files.

**Do you want your identity to be public for this peer review?** For information about this choice, including consent withdrawal, please see our Privacy Policy .

Reviewer #1: **Yes: ** Rashmi Agarwalla

Reviewer #2: No

---

## [Author Response · Author response to Decision Letter 1]

11 Jan 2025

Point-By-Point Response

We are deeply grateful to the editor and all reviewers for their invaluable contributions to enhancing the quality and clarity of our manuscript. The detailed and insightful feedback, particularly on the methodology and statistical analysis sections, has significantly improved the rigor and comprehensiveness of our study. Your thorough review and constructive comments have been instrumental in refining our work.

In the following pages, we provide a point-by-point response to all questions and suggestions raised by the editor and reviewers. All changes made in the manuscript are highlighted in blue.

Reviewer ≠1

1. The Table 1 can be made more concise and data can be grouped in 5ys and shown

Author response: We appreciate the reviewer’s suggestion to make Table 1 more concise by grouping the data into 5-year intervals. To address this, we have revised Table 1 by reorganizing the data accordingly. The updated table provides a more streamlined presentation, while maintaining the clarity and integrity of the data. We believe this modification enhances the readability and utility of the table for the readers.

2. Recommendations and targeted interventions need to be specified more as to what targeted interventions can be done age wise or gender wise.

Author response: Thank you for your feedback. We have added a Recommendations and Targeted Interventions section after the conclusion to specify more clearly the age- and gender-targeted strategies. These interventions now include specific approaches for women, adolescent girls and young women, and older adults, as well as region-specific strategies for high-burden areas like Gambela. The revised section also highlights targeted prevention efforts for high-risk groups and expands on the need for equitable access to HIV treatment for all populations, including maternal health services for women and pediatric ART for children and adolescents.

Reviewer #2

1. The manuscript appears to be technically sound. The results regarding the trends in HIV/AIDS in Ethiopia are supported by data from the Global Burden of Disease Study, which was analyzed over a significant time span. The methodology seems appropriate, and the findings are strengthened by the use of gender, regional, and age-standardized data. The conclusions are well-supported by the data, and the manuscript offers useful suggestions for addressing persistent HIV/AIDS-related challenges in Ethiopia.

Author response: Thank you for your positive feedback. We are glad to hear that you find the manuscript technically sound and that the use of data from the Global Burden of Disease Study, along with the gender, regional, and age-standardized analysis, has strengthened the finding.

2. The statistical analysis appears to be appropriate and rigorous, with sound methods used for analyzing the trends.

Author response: Thank you for your feedback. We appreciate your recognition of the statistical analysis as appropriate and rigorous. We are confident that the methods used to analyze the trends are robust and have helped ensure the reliability and validity of the findings. We will continue to ensure that the statistical approach remains transparent and well-documented throughout the manuscript.

3. The manuscript is generally clear and well-structured, with a logical flow from introduction to methods, results, and conclusion. The language is mostly correct and appropriate for a scientific manuscript. However, it might benefit from further elaboration on terms like "disability-adjusted life years (DALYs)" to ensure clarity, especially for readers who may not be familiar with the concept. Additionally, I suggest standardizing the use of abbreviations throughout the manuscript for consistency and readability, particularly for terms like DALYs, GBD, ART, and others. This will enhance the clarity and uniformity of the manuscript.

Author response: Thank you for your valuable feedback. In response to your suggestion, we have added further elaboration on terms like "disability-adjusted life years (DALYs)" in the methodology section to ensure clarity, particularly for readers who may not be familiar with the concept. Additionally, we have standardized the use of abbreviations such as DALYs, GBD, ART, and others throughout the manuscript to improve consistency and readability. We believe these revisions enhance the overall clarity and uniformity of the manuscript.

---

## [Editor Report · Decision Letter 1]

28 Feb 2025

The Burden of HIV/AIDS in Ethiopia: Unveiling 30 Years of Trends in Incidence, Mortality, and Disability—Insights from the Global Burden of Disease Study (1990–2021).

PONE-D-24-52871R1

Dear Dr. Tegene Atamenta Kitaw,

We’re pleased to inform you that your manuscript has been judged scientifically suitable for publication and will be formally accepted for publication once it meets all outstanding technical requirements.

Kind regards,

Deepak Dhamnetiya, MD

Academic Editor

PLOS ONE

---

## [Editor Report · Acceptance letter]

PONE-D-24-52871R1

PLOS ONE

Dear Dr. Kitaw,

I'm pleased to inform you that your manuscript has been deemed suitable for publication in PLOS ONE. Congratulations! Your manuscript is now being handed over to our production team.

Kind regards,

on behalf of

Dr. Deepak Dhamnetiya

Academic Editor

PLOS ONE